# First Case Reports of Nontuberculous Mycobacterial (NTM) Lung Disease in Ecuador: Important Lessons to Learn

**DOI:** 10.3390/pathogens12040507

**Published:** 2023-03-23

**Authors:** Gustavo Echeverria, Veronica Rueda, Wilson Espinoza, Carlos Rosero, Martín J. Zumárraga, Jacobus H. de Waard

**Affiliations:** 1Instituto de Investigación en Zoonosis-CIZ, Universidad Central del Ecuador, Quito 170518, Ecuador; 2Programa de Doctorado, Facultad de Ciencias Veterinarias, Universidad de Buenos Aires, Buenos Aires C1063ACV, Argentina; 3División Investigación y Desarrollo, BioGENA, Quito 170509, Ecuador; 4Departamento de Tuberculosis, Hospital de Especialidades Eugenio Espejo, Quito 170401, Ecuador; 5Instituto de Agrobiotecnología y Biología Molecular, IABIMO, INTA-CONICET, Buenos Aires C1063ACV, Argentina; 6Departamento de Tuberculosis, Instituto de Biomedicina “Jacinto Convit”, Universidad Central de Venezuela, Caracas 1010, Venezuela; 7One Health Research Group, Facultad de Ciencias de la Salud, Universidad de Las Américas (UDLA), Quito 180602, Ecuador

**Keywords:** nontuberculous mycobacteria (NTM), pulmonary infection, tuberculosis, MAC infection, Ecuador

## Abstract

Nontuberculous mycobacteria (NTM) lung infections are often misdiagnosed as tuberculosis, which can lead to ineffective antibiotic treatments. In this report, we present three cases of NTM lung infections in Ecuador that were initially diagnosed and treated as tuberculosis based on the results of sputum smear microscopy. The patients, all male, included two immunocompetent individuals and one HIV-positive subject. Unfortunately, sputum culture was not initiated until late in the course of the disease and the cause of the lung infection, *Mycobacterium avium* complex (MAC), was only identified after the patients had either passed away or were lost to follow-up. These cases are the first documented cases of NTM lung infections in the English medical literature from Ecuador. We emphasize the importance of accurate diagnosis of NTM infections by culture and identification to species level. Sputum smear staining alone cannot differentiate between mycobacterial species, which can lead to misidentification and ineffective treatments. Additionally, reporting NTM pulmonary disease as a notifiable disease to national TB control programs is recommended to obtain accurate prevalence data. These data are critical in determining the importance of this public health problem and the necessary actions needed to address it.

## 1. Introduction

The genus *Mycobacterium* includes more than 200 species including the main human pathogens of the *Mycobacterium tuberculosis* complex and *M. leprae*, as well as numerous other environmental species, nontuberculous mycobacteria (NTM), that can be found in water and soil. These NTM are occasionally responsible for opportunistic infections in humans and animals [1,2]. Although pulmonary infections caused by mycobacteria are most commonly caused by one of the members of the *M. tuberculosis* complex, over the last decades, NTM has become increasingly prevalent as the causative agent of pulmonary infections [3], and in some countries, NTM pulmonary infection has become more important than infection by a member of the *M. tuberculosis* complex. An example is Canada where during a study period of 6 years, the isolation prevalence of NTM was approximately twice as great as the rates of pulmonary tuberculosis cases [4]. Another example of a region with a high registered prevalence of NTM isolation is Shanghai, China, where in the year 2008, in a key laboratory for the diagnosis of tuberculosis, 6.38% of the mycobacterial clinical isolates from sputum samples were NTM. [5] In the United States of America (USA), from 2008 to 2015, the yearly prevalence estimates for NTM lung disease in people younger than 65 years of age increased from 2.87 to 4.10 per 100,000 people and in those aged 65 years or older from 30.27 to 47.48 per 100,000 people. [6] Furthermore, in Ethiopia, in 2017, among 3317 samples in the reference laboratory with mycobacterial culture results, 7.3% were NTM. [7]

In addition to immunosuppression (such as HIV infection or the use of immunosuppressant drugs), older age, being female, and previous TB treatment are known risk factors for NTM lung infections [8]. Another significant risk factor for NTM lung disease is bronchiectasis, and a recent meta-analysis estimated that the global prevalence of NTM infection in adults with non-cystic fibrosis bronchiectasis is approximately 10%, with considerable variations primarily due to geographical location [9]. Furthermore, people with cystic fibrosis are at an increased risk of lung infection with NTM. According to a review, the overall prevalence of NTM infection in cystic fibrosis patients is 7.9% [10].

As the symptoms, signs, and radiological characteristics of NTM lung infection are similar to those of tuberculosis (TB), most NTM cases are diagnosed based on the suspicion of tuberculosis. In many developing countries, including Ecuador, sputum smear microscopy is the primary method for diagnosing tuberculosis, and culturing of sputum samples is not routinely performed. However, Ziehl–Neelsen (ZN) staining is not able to differentiate between mycobacterial species, and as the initial laboratory diagnosis is typically “smear positive”, an NTM infection is often misdiagnosed as tuberculosis. Consequently, most patients with an NTM pulmonary infection will receive anti-tuberculosis treatment [11]. However, many of the commonly identified NTM species, particularly the rapid growers, of the genus *Mycobacterium* such as *M. abscessus, M. chelonae*, and *M. fortuitum*, are resistant to most or all of the first- and second-line drugs used to treat tuberculosis [12,13]. As a result, patients with NTM infections who do not respond to anti-TB treatment may be erroneously classified as having multidrug-resistant tuberculosis (MDR-TB), as has been reported in Iran [14].

The epidemiology of pulmonary NTM disease in Ecuador remains unknown, and there are no published scientific reports on this topic. Furthermore, as NTM infection is not a notifiable disease, as in most countries, there are also no official government reports on the incidence or prevalence of NTM lung infections.

In this study, we aimed to identify NTM isolates from sputum specimens collected at a single hospital in Quito, Ecuador, where culture for tuberculosis diagnosis was introduced in 2017. We present three patients who were initially diagnosed with TB in that same year but later found to have NTM lung disease. Our study highlights the critical role of culturing sputum samples and identifying the isolated *Mycobacterium* species to the species level in the diagnosis and treatment of respiratory infections. This enables physicians to provide an accurate diagnosis, successful treatment plan, and a cure for the patient while avoiding unnecessary anti-TB treatments, confusion with MDR-TB, extensive lung damage, or even death. 

## 2. Materials and Methods

### 2.1. Selection of Patients with NTM Pulmonary Disease

In 2017, the Eugenio Espejo hospital, a 416-bed referral hospital in Quito, sputum culture was implemented to enhance the accuracy of sputum examination and improve diagnostic yield, particularly for cases where Ziehl–Neelsen staining (ZN) is not sufficiently sensitive for diagnosis. Sputum samples, after a decontamination step with 4% NaOH, were cultured on Löwenstein–Jensen medium at 37 °C. Culture positive sputum samples for acid-fast bacilli (AFB) of that year, 2017, were further examined for the presence of NTM isolates. The NTM isolates were initially identified by phenotypic characteristics including colony morphology, pigment, and growth rate.

### 2.2. Identification of the NTM Isolates

The identity of NTM isolates was further confirmed using a PCR-restriction enzyme pattern analysis (PRA) method based on the hsp65 gene [15]. Species identification was then confirmed using 16S rRNA sequence analysis through BLAST® (NCBI services, Bethesda, MD, USA), with the following primers used for amplification: A1F (CTGGCTCAGGACGAACGCTG) and 54R (TCTAGTCTGCCCGTATTCGCCC) [16]. Since the isolates could not be confidently identified via 16S rRNA sequencing and nBLAST search analysis only, a phylogenetic method was employed using the 16S rDNA sequences [17].

### 2.3. Patients Records

The hospital provided a copy of the medical records for each patient with an NTM isolate, which included details about the patient’s medical history, clinical findings, and diagnostic test results. The data were collected by the responsible medical practitioner (CR), and the copied records were destroyed after data collection.

## 3. Results

In 2017, the hospital laboratory received 1011 sputum samples for TB diagnosis. As sputum samples are cultured only upon physician’s request, only 260 samples were cultured and the remaining samples were examined solely by smear microscopy. Of those cultured sputum samples, 26 (10%) were culture positive for acid-fast bacilli (AFB). Notably, seven of these positive cultures were negative for ZN staining and AFB smear microscopy, showing a sensitivity for smear microscopy of 73% in comparison with culturing of sputum samples for AFB isolates. A breakdown of the results can be found in Table 1. Four isolates displayed characteristics of NTM such as a smooth colony morphology (three isolates) and a fast growth rate (one isolate). The cultures were pure, containing a single species of organism. To confirm the identity of these isolates, further identification was conducted using the PRA technique and sequencing of the 16 S rRNA gene.

From three patients, four NTM strains were isolated, comprising of one rapid-growing, identified as *M. phlei*, and three slow-growing mycobacteria, all identified as *M. chimaera/M. intracellulare*. Medical records for each patient were available for further analysis. According to the diagnostic criteria for NTM lung disease as per the ATS/ERS/ESCMID/IDSA Clinical Practice Guidelines [18], all three patients had pulmonary symptoms and had tested positive for ZN stain in at least two separate expectorated sputum samples during the course of their disease. Additionally, chest radiographs indicated nodular or cavitary opacities in all three cases.

**Patient 1:** was a 57-year-old immunocompetent male, without underlying diseases, from Quito. In 2007, he was diagnosed with pulmonary tuberculosis and treated for 6 months with standard TB treatment, resulting in a cure. In 2013, he was re-admitted with positive ZN sputum smear staining and underwent another 9-month course of standard TB treatment. In 2015, he presented again with pulmonary symptoms, fever, and weight loss. Despite treatment, he did not improve, and his disease was cataloged as resistant TB. A sputum sample was cultured and the drug-resistant isolate was identified in a reference laboratory as compatible with *M. phlei/M. celatum/M. chubuense/M. chlorophenolicum*. The patient was treated with clarithromycin, trimethoprim–sulfamethoxazole, rifampicin, ethambutol, and doxycycline, but unfortunately passed away in 2017. GeneXpert MTB PCR was not performed on the sputum samples. Subsequent re-identification with the PRA technique and 16S rRNA sequencing identified the strain as *M. phlei*, which is a rare infection with fewer than 10 reported cases in the literature [19]. After his death, two other sputum cultures became available and both were identified as *M. chimaera / M. intracellulare* through molecular identification (see Appendix A).

**Patient 2**: was a 30-year-old HIV-positive male from Quito who was hospitalized in 2017 with a fever, severe headache, and respiratory problems. His cerebrospinal fluid was suggestive of TB, showing a characteristic profile with predominantly lymphocytes, elevated protein, and low glucose. TB treatment was initiated, and samples of urine, feces, and sputum were sent for tuberculosis culture. While AFB were observed in sputum smear microscopy, GeneXpert did not detect the DNA of *M. tuberculosis*. After 10 days, the patient was discharged from the hospital with antiretroviral therapy but did not return for a check-up. The final diagnosis for this subject was disseminated NTM disease, including involvement of the lungs. CD4 count of the patient was not recorded in his clinical history, but a study in Mexico showed that disseminated Mycobacterium avium complex (MAC) infection occurs in 20–40% of patients with <50 CD4/mm^3^ [20]. Ten days after being discharged, two different sputum cultures grew AFB, and subsequent PRA and 16S rRNA sequencing identified the isolates as *M. chimaera/M. intracellulare* (see Appendix A).

**Patient 3:** was a 46-year-old immunocompetent male from Quito who was admitted to the hospital’s Intensive Care Unit in February 2016 with symptoms of general malaise, cough with expectoration, and acute respiratory failure. He had been diagnosed with bronchiectasis two years prior. The patient was started on treatment with ciprofloxacin. Two sputum samples were collected and both tested positive for AFB smear microscopy, suggestive for a diagnosis of pulmonary tuberculosis. GeneXpert MTB PCR was not performed on the sputum samples, and a standard TB treatment was initiated. Despite being hospitalized for 30 days, the patient did not survive. Ten days after his death, two cultures were obtained and identified with PRA pattern and 16 S rRNA sequencing as *M. chimaera/M. intracellulare*. (see Appendix A).

## 4. Discussion

In 2017, three patients in the Quito hospital were misdiagnosed with tuberculosis but were later found to have a lung infection caused by a species of the *Mycobacterium avium* complex (MAC). Unfortunately, the true cause of the respiratory infection was only identified after the patients had already been treated with an anti-TB regime. Concerning patient 1, who had a culture positive for *M. phlei*, and two other cultures with an MAC strain, we suppose that the first isolate was a temporary colonization with this species, as *M. phlei* is generally a nonpathogenic mycobacterium, and this species was not isolated in the two subsequent sputum cultures [19].

Two patients died and the hospital lost track of the third patient. It is unclear whether the MAC infection was the sole cause of death or if other underlying diseases were involved. The drugs typically used for MAC infections include a macrolide (such as clarithromycin or azithromycin), ethambutol, and a rifamycin (such as rifabutin or rifampin). Most first-line anti-TB drugs have 10 to 100 times less in vitro activity against MAC isolates than against *M. tuberculosis* [18]. Two studies evaluating the long-term treatment response to anti-TB medications against MAC infection showed that relapses after medical therapy with anti-TB treatment regimens were common and only approximately 50% of the patients had a long-term favorable response. [18]. 

In this small study conducted in Quito, the prevalence of NTM lung infection, as determined by culture results, was approximately 10%. However, there is a lack of previous reports on NTM lung infection in Ecuador in both the medical literature and hospital and national TB program records. This may be due to the fact that the majority of TB cases in Ecuador are diagnosed solely through AFB smear staining and microscopy, without confirmation through culture, resulting in NTM infections being missed. GeneXpert is not used routinely.

NTM lung disease is more commonly reported in industrialized countries, with limited information available from less developed regions. In Latin America, Argentina has reported a prevalence of approximately 5% of NTM cases among 23,624 clinical specimens investigated during the period of 2004–2010 [21]. Brazil has been the focus of most reports on NTM lung infections, with *M. avium complex* (MAC), *M. kansasii*, and *M. abscessus* being the principal isolated species in the state of Rio Grande do Sul [22]. Another study from Brazil analyzed 1812 respiratory secretion samples and yielded 75 NTM isolates, with *M. abscessus* and *M. avium* being the most important species [23]. The only publication in South America that investigated the prevalence of NTM in cystic fibrosis patients also comes from Brazil. Of 117 patients with respiratory samples cultured for NTM research, this study found seven patients (6%) with at least one positive result for NTM [24]. In Mexico, 16 NTM strains were isolated from 99 pulmonary samples, with *M. avium* and *M. intracellulare* being the most important mycobacterial species [25]. In Colombia, a study of HIV patients showed that *M. avium* was isolated in 4.5% of the study group of 265 patients [26]. Despite an extensive search in English scientific literature, we could not find any other research reports of countries in South America on the prevalence of NTM lung infections. While Venezuela and Ecuador have reported multiple cases and outbreaks of skin and soft tissue infections caused by NTM, there is no documentation of any NTM lung disease cases in these countries [27,28,29,30,31,32,33].

With regards to culturing for the isolation of mycobacteria, we consider this an essential tool that should always be present in a tuberculosis laboratory. Culturing is necessary to confirm a tuberculosis diagnosis and to identify NTM strains. Molecular methods, such as the GeneXpert MTB/RIF assay, can be used for the initial diagnosis of TB but cannot replace conventional culture techniques as it concerns an NTM infection. However, when one suspects an NTM infection or when an AFB smear-positive patient has a negative GeneXpert result, culture should resolve this clinical dilemma. Moreover, AFB smear-negative sputum should be cultured as it has been shown that GeneXpert performs suboptimally compared with the culture of mycobacteria [34,35]. Our small study clearly showed that culture improved the diagnostic yield by approximately 36% (26 patients with a positive culture for mycobacteria versus 19 smear-positive patients). Fast and cheap culture methods are available for culturing mycobacteria, such as the Kudoh swab method [36]. This method is extensively used in Japan and has been evaluated in several publications, and the sensitivity is comparable to that of the standard Petroff culture procedure. Moreover, this method is cheap, requires no centrifugation step, and takes only 4–5 min per sputum sample [37]. We acknowledge that culture for mycobacteria is impeded by slow growth and that a conclusive diagnosis of NTM infection can take several weeks. Over the past two decades, commercial and in-house molecular methods have emerged as dependable and rapid substitutes for laboratory diagnostics of NTM in clinical samples [38,39,40]. However, these methods require expensive equipment and running costs, making them unattainable for most laboratories in developing countries, and therefore not yet a feasible option for faster turnover.

## 5. Conclusions

Overall, this report emphasizes the need for accurate diagnosis and reporting of NTM infections. The burden of NTM lung disease is increasing worldwide and the infection shares clinical and radiological features with TB [41,42]. Moreover, as shown in this publication, its diagnosis is frequently delayed and confused with TB. Our study showed that a final diagnosis is not established with only an AFB sputum test, and a sputum culture is recommended for all patients with suspected NTM lung disease.

Currently, NTM disease is not nationally notifiable in Ecuador or in most other countries worldwide. This is because the transmission of NTM between people is believed to be extremely rare, and thus, NTM disease is not typically considered a public health problem. However, based on the increasing number of cases reported in the medical literature, we recommend that NTM disease be registered and monitored. We also suggest that culture be performed for all AFB-negative patients or AFB-positive patients whose diagnoses are not confirmed with molecular detection, such as the GeneXpert assay. Additionally, since most countries lack data on the prevalence of NTM lung disease, we recommend operational studies to determine prevalence in different geographical regions of the country. It is well known that species distribution and frequency of infection differ by continent and by countries within those continents [43,44].

## 6. Limitations of This Study

Our study was limited to the culture results of a single hospital in Quito, which is one of the few hospitals in Ecuador that grows mycobacteria when a case of tuberculosis is suspected. Additionally, only 25% of sputum samples were cultured during the study year, and the study period was short, lasting only one year. Furthermore, only 4 out of 26 positive cultures from the hospital laboratory in 2017 were identified as NTM isolates, with the rest being considered *M. tuberculosis* isolates based on visual identification, which is known to be unreliable. As a result, it is possible that cases of NTM lung infections were missed

## Figures and Tables

**Table 1 pathogens-12-00507-t001:** The number of received sputum samples processed for tuberculosis diagnosis in the laboratory in 2017. The table shows the number of AFB sputum tests, sputum culture, and isolates for identification.

Sputum Samples (n)	AFB Positive (n)	Sputum Culture (n)	AFB Positive Culture Positive (n)	AFB Negative Culture Positive (n)	Isolates for Identification (n)	NTM (n)
1011	37	260	19	7	4	4

## Data Availability

Not applicable.

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
