# Peer review of "First Case Reports of Nontuberculous Mycobacterial (NTM) Lung Disease in Ecuador: Important Lessons to Learn"

_pathogens, 2023, doi:10.3390/pathogens12040507_

Round 1
Reviewer 1 Report
These authors successfully highlight the issue of NTM infections among persons with suspected TB, and the clinical consequences of lack of species identification for positive AFB smears in this group. The background is thorough, and the data and conclusions are well presented.
My only question relates to why were not all 1,000 sputum samples cultured? was this due to lack of resources? Is this standard hospital procedure? This point is not clear. Please clarify
Author Response
Dear reviewer
Thank you for carefully reviewing our manuscript and providing us with constructive feedback. We have taken your comments and suggestions into consideration and have provided our responses in red under each of your questions and remarks.
"My only question relates to why were not all 1,000 sputum samples cultured? was this due to lack of resources? Is this standard hospital procedure? This point is not clear. Please clarify"
Sputum samples were only cultured on request of the physician. We added this to the text. See line 126-127.
Sincerely,
Jacobus H. de Waard
Reviewer 2 Report
The authors present three cases of NTM infection in Ecuador and discuss the importance oft he differentiation between MTB and NTM within the diagostic workflow from the perspective of a developing country.
Comments:
Line 31: replace instead of replaced
Line 45: Currently, more than 188 species have been described.
Line 50: M. tuberculosis instead of Mycobacterium tuberculosis
Line 78: Mycobacterium avium, Mycobacterium kansasii, Mycobacterium abscessus (full name at the frist mention)
Line 82: Mycobacterium intracellulare
Line 109: Mycobacterium chelonae, Mycobacterium fortuitum
Line 113ff: PCR vs culture for MTB and NTM detection. To round off the topic, I would recommend discussing the options for direct PCR detection of NTM from clinical materials (commercial systems, in-house solutions). Especially from smear positive samples, their sensitivity is very good.
Line 125 Materials and Methods. Please describe in more detail how the cultural examination was carried out (decontamination, liquid medium, solid media, time and temperature of incubation). And how was within the diagnostic process the possibility of mixed cultures of MTB and NTM adressed? This is an important issue for patients with previous TB treatment.
Line 141: Has an ethics proposal been approved and was patient consent obtained?
Line 160: Please use the latest guidelines form 2020 (Daley et al).
Line 183: Patient 2: Was the patient under HIV treatment? What was his CD4 count?
Line 193: Patient 3: Was Xpert MTB PCR performed on the sputum samples? Was the patient immuonocompromised? HIV?
Author Response
Dear reviewer
Thank you for carefully reviewing our manuscript and providing us with constructive feedback. We have taken your comments and suggestions into consideration and have provided our responses in red under each of your questions and remarks.
Sincerely,
Jacobus H. de Waard
Comments:
Line 31: replace instead of replaced
Done
Line 45: Currently, more than 188 species have been described.
188 was changed in more than 200.
Line 50: M. tuberculosis instead of Mycobacterium tuberculosis
Done
Line 78: Mycobacterium avium, Mycobacterium kansasii, Mycobacterium abscessus (full name at the first mention) Line 82: Mycobacterium intracellulare Line 109: Mycobacterium chelonae, Mycobacterium fortuitum
We avoided now the repeated use of the genus name Mycobacterium by adding the words; “of the genus Mycobacterium” in line 83
Line 113ff: PCR vs culture for MTB and NTM detection. To round off the topic, I would recommend discussing the options for direct PCR detection of NTM from clinical materials (commercial systems, in-house solutions). Especially from smear positive samples, their sensitivity is very good.
We added this discussion to the discussion section line 249-255
Line 125 Materials and Methods. Please describe in more detail how the cultural examination was carried out (decontamination, liquid medium, solid media, time and temperature of incubation). And how was within the diagnostic process the possibility of mixed cultures of MTB and NTM addressed? This is an important issue for patients with previous TB treatment.
On solid medium it is relatively easy to differentiate different colony morphology.
We added the following to the material and method section; Sputum samples, after a decontamination step with 4% NaOH were cultured on Löwenstein–Jensen medium at 37°C.
We also added to the results section that pure cultures were isolated. See line 135-136
Line 141: Has an ethics proposal been approved and was patient consent obtained?
This has been addressed in the In the Institutional Review Board Statement in line 297-305
Line 160: Please use the latest guidelines form 2020 (Daley et al). Done.
Line 183: Patient 2: Was the patient under HIV treatment? What was his CD4 count? antiretroviral therapy. We added the following information to the patient’s history: with antiretroviral therapy, The CD4 count was not registered in his clinical history. See line
Line 193: Patient 3: Was Xpert MTB PCR performed on the sputum samples? Was the patient immuonocompromised? HIV?
We added to all the patients histories if GeneXpert was used, underlying diseases and HIV status.
Reviewer 3 Report
Overall comments
I read with interest the paper by Echeverría et al. who describes three cases of NTM infection in Ecuador and highlights the importance of culturing as diagnostics to distinguish between NTM and TB. There is definetly lack of NTM data, as also stated by the authors, in many settings, particularly in low-income countries. Consequently, I think the study is warrented. However, the study should be improved in terms of structure but also English phrasing and syntax. E.g., tuberculosis should not be capitalized. Repetitions should be avoided and a clear focus throughout the manuscript is needed. The introduction is very long compared to the very brief results and discussion section. The introduction needs to be shortened and be more focused on the aim of the study. Also, some parts could be moved to the discussion or deleted. For instance, in lines 113-114 you write that culturing is indispensable and then again in lines 119-120 you state that with your study “…show the need for culturing sputum”. Your focus should be the lack of epidemiological and clinical data from Ecuador (and South American in general). You don’t need to describe every detail about NTM infections. In addition, your aim in lines 119 and 124 could be sharper. “E.g. In this case series, we describe the clinical / microbiological characteristics of three patients with NTM-PD in Ecuador, emphasizing/showing…” and so on. Finally, references are no appropriately placed and important literature may be missing. Below are some specific comments.
Abstract
Line 25-27: Loss of the patient, you mean lost to follow-up or? And was this the case for all three patients and that they were considered to have TB at first?
Line 27-28: Important to identify subspecies for all mycobacteria (including NTM) and not only M. tuberculosis. Please paraphrase sentence.
I think the abstract is mostly about your opinion about NTM and mycobacteria diagnostics (8 out of 12 lines), which I agree with, but I think you should describe the cases findings a bit more, since it is a case series you’re presenting.
Introduction
Line 45-49: The first sentence is a bit hard to read. It should be paraphrased or split up.
Line 49-53: https://doi.org/10.1016/j.ijid.2022.10.013 and doi.org/10.1016/j.ccm.2014.10.002 are systematic reviews that could be used as references to support your statements.
Line 58: 6.38% compared to what? How many did it used to be? Or how many isolates are M. tuberculosis?
Materials and methods
Line 127: But what is the study period? 2017? How were patients identified? Please elaborate on the design of the study and the inclusion of samples / patients.
Ethical considerations? Did patients consent to the study?
Results
Line 148-149: What happened to the 751 other samples? Were these only examined with microscopy? So, among 260 samples examined for TB and NTM, 10% were positive for NTM? That is really a high rate of positive isolates.
Line 151: How was diagnostic yield defined and calculated? And why do you state the TB diagnostic yield and not NTM?
Line 162-164: No need to state this as you just stated the patients fulfilled ATS/IDSA criteria. If you want to explain the criteria, I think that should have been done in the methods section.
Patient 1: No underlying disease?
Patient 2: You state you present data from NTM lung infection in your study, but to be honest, case no. two has disseminated disease if he is HIV positive and you considered the AFB in the cerebrospinal fluid to be NTM.
Patient 3: Ciprofloxacin, ranitidine, metoclopramide, and furosemide for bronchiectasis? How does that make sense?
Line 212: “…according to the literature” but you did not provide any references
Lines 213-216: No references
Discussion
How prevalent is CNS involvement in patients with MAC and HIV?
Line 226-229: But GeneXpert is not used routinely?
Conclusion
I think the cocnlusion should be shoretned and be based on your study findings of this study and only shortly on your opinion / other studies.
Table and figures
The supplementary figure is in a very poor resolution and hard to evaluate
Other references
During a quick PubMed search, I found the following studies with data on pulmonary NTM, from South America (mainly Brazil) that was not referenced or mentioned:
https://pubmed.ncbi.nlm.nih.gov/32474494/
https://pubmed.ncbi.nlm.nih.gov/31038648/
https://pubmed.ncbi.nlm.nih.gov/29846573/ (CF patients)
https://pubmed.ncbi.nlm.nih.gov/29791556/
https://pubmed.ncbi.nlm.nih.gov/19099102/
Author Response
Dear reviewer
Thank you for carefully reviewing our manuscript and providing us with constructive feedback. We have taken your comments and suggestions into consideration and have provided our responses in red under each of your questions and remarks.
Sincerely,
Jacobus H. de Waard
Overall comments
I read with interest the paper by Echeverría et al. who describes three cases of NTM infection in Ecuador and highlights the importance of culturing as diagnostics to distinguish between NTM and TB. There is definetly lack of NTM data, as also stated by the authors, in many settings, particularly in low-income countries. Consequently, I think the study is warrented. However, the study should be improved in terms of structure but also English phrasing and syntax. E.g., tuberculosis should not be capitalized. Repetitions should be avoided and a clear focus throughout the manuscript is needed. The introduction is very long compared to the very brief results and discussion section. The introduction needs to be shortened and be more focused on the aim of the study. Also, some parts could be moved to the discussion or deleted. For instance, in lines 113-114 you write that culturing is indispensable and then again in lines 119-120 you state that with your study “…show the need for culturing sputum”. Your focus should be the lack of epidemiological and clinical data from Ecuador (and South American in general). You don’t need to describe every detail about NTM infections. In addition, your aim in lines 119 and 124 could be sharper. “E.g. In this case series, we describe the clinical / microbiological characteristics of three patients with NTM-PD in Ecuador, emphasizing/showing…” and so on. Finally, references are no appropriately placed and important literature may be missing. Below are some specific comments.
We appreciate your valuable suggestions and have made significant revisions to our manuscript accordingly. We have removed certain sections from the introduction and discussion that were deemed irrelevant. Furthermore, we have meticulously reviewed and linked our references to the corresponding text to ensure accuracy. In addition, we have added relevant references to strengthen our arguments.
Abstract
Line 25-27: Loss of the patient, you mean lost to follow-up or? And was this the case for all three patients and that they were considered to have TB at first?
We have updated our manuscript to clarify that all three cases were initially diagnosed as TB cases, and unfortunately, two patients passed away while one was lost to follow-up. We hope that this clarification provides a better understanding of our findings and conclusions.
Line 27-28: Important to identify subspecies for all mycobacteria (including NTM) and not only M. tuberculosis. Please paraphrase sentence. Done.
I think the abstract is mostly about your opinion about NTM and mycobacteria diagnostics (8 out of 12 lines), which I agree with, but I think you should describe the cases findings a bit more, since it is a case series you’re presenting. Done.
Introduction
Line 45-49: The first sentence is a bit hard to read. It should be paraphrased or split up.
Done. We split the phrase up.
Line 49-53: https://doi.org/10.1016/j.ijid.2022.10.013 and doi.org/10.1016/j.ccm.2014.10.002 are systematic reviews that could be used as references to support your statements.
We use now both reviews as a reference.
Line 58: 6.38% compared to what? How many did it used to be? Or how many isolates are M. tuberculosis?
We report now: key laboratory for the diagnosis of tuberculosis, 6.38% of the mycobacterial clinical isolates from sputum samples were NTM.
Materials and methods
Line 127: But what is the study period? 2017? How were patients identified? Please elaborate on the design of the study and the inclusion of samples / patients.
We elaborated this part. Line 104-119
Ethical considerations? Did patients consent to the study?
Ethical considerations can be found in line 298-305. Institutional Review Board Statement.
Results
Line 148-149: What happened to the 751 other samples? Were these only examined with microscopy? So, among 260 samples examined for TB and NTM, 10% were positive for NTM? That is really a high rate of positive isolates.
We clearly state now that the other sputum samples were only examined by microscopy. Line 127-129
Line 151: How was diagnostic yield defined and calculated? And why do you state the TB diagnostic yield and not NTM?
Diagnostic yield was calculated for sputum samples with an AFB isolate. See line 132-133
Line 162-164: No need to state this as you just stated the patients fulfilled ATS/IDSA criteria. If you want to explain the criteria, I think that should have been done in the methods section.
I apologize for any confusion. To clarify, our study focused exclusively on non-tuberculous mycobacteria (NTM) isolates, which we selected through our material and method selection process. In the results section, we will make sure to specify that the cultures were obtained from subjects with NTM disease to provide further context for our findings.
Patient 1: No underlying disease? No underlying disease. We added this information to the case 1.
Patient 2: You state you present data from NTM lung infection in your study, but to be honest, case no. two has disseminated disease if he is HIV positive and you considered the AFB in the cerebrospinal fluid to be NTM. Yes, you are right. We changed this in: The final diagnosis for this subject was disseminated NTM disease, including involvement of the lungs. Line 171-172.
Patient 3: Ciprofloxacin, ranitidine, metoclopramide, and furosemide for bronchiectasis? How does that make sense?
You are right and this makes no sense. However, this was noted in the clinical history and prescribed for other symptoms: We removed the medicines ranitidine, metoclopramide, and furosemide from the text.
Line 212: “…according to the literature” but you did not provide any references. Reference 18. It is now there.
Lines 213-216: No references. Reference 18. It is now there.
Discussion
How prevalent is CNS involvement in patients with MAC and HIV? See line 174 – 178. We added a reference.
Line 226-229: But GeneXpert is not used routinely? This information is now in line 212-214. We added this observation.
Conclusion
I think the conclusion should be shortened and be based on your study findings of this study and only shortly on your opinion / other studies. We shortened the conclusions.
Table and figures
The supplementary figure is in a very poor resolution and hard to evaluate. We improved the resolution.
Other references
During a quick PubMed search, I found the following studies with data on pulmonary NTM, from South America (mainly Brazil) that was not referenced or mentioned:
Most of these studies are out of the scope of this article and deal with subjects as treatment. but we mention now the following reference in line 223-226
https://pubmed.ncbi.nlm.nih.gov/29846573/ (CF patients)
Aiello TB, Levy CE, Zaccariotto TR, Paschoal IA, Pereira MC, Nolasco da Silva MT, Ribeiro JD, Ribeiro AF, Toro AADC, Mauch RM. Prevalence and clinical outcomes of nontuberculous mycobacteria in a Brazilian cystic fibrosis reference center. Pathog Dis. 2018 Jul 1;76(5). doi: 10.1093/femspd/fty051.
----------------------------------------------------------------------------------------------
Thank you very much for critical reading. The article improved considerably!
Jacobus H. de Waard. Corresponding author.
Round 2
Reviewer 3 Report
Dear editorial board and authors,
From my perspective, the authors have now answered all my concerns and the manuscript has improved significantly. I suggest acceptance.
Best regards,